Serotonin 5-HT1A receptor binding and self-transcendence in healthy control subjects—a replication study using Bayesian hypothesis testing

Griffioen Gina gina.griffioen@ki.se ginagriffioen@gmail.com 1 2
Matheson Granville J. 1
Cervenka Simon 1
Farde Lars 1 3
Borg Jacqueline 1
1 Centre for Psychiatry Research, Department of Clinical Neuroscience, Karolinska Institutet and Stockholm County Council , Stockholm , Sweden
2 Capio Psykiatri Stockholm , Stockholm , Sweden
3 Personalised Healthcare and Biomarkers, AstraZeneca PET Science Centre, Karolinska Institutet , Sweden
Black Kevin
Electronic publication date: 2018 Nov 16
Publication date: 2018
Volume: 6
Electronic Location ID: e5790
Received 2018 Jan 18; Accepted 2018 Sep 19
Copyright: ©2018 Griffioen et al.
Copyright year: 2018
Copyright holder: Griffioen et al.
License: This is an open access article distributed under the terms of the Creative Commons Attribution License, which permits unrestricted use, distribution, reproduction and adaptation in any medium and for any purpose provided that it is properly attributed. For attribution, the original author(s), title, publication source (PeerJ) and either DOI or URL of the article must be cited.
License URL: https://creativecommons.org/licenses/by/4.0/

Keywords: Serotonin, Bayes theorem, Replicability, Spirituality, Self-transcendence, 5-HT1A, Positron Emission Tomography

Funding: Swedish Research Council 2015-02398 523-2014-3467 This study was supported by the Swedish Research Council (2015-02398 (Lars Farde); 523-2014-3467 (Simon Cervenka)). There was no additional external funding received for this study. The funders had no role in study design, data collection and analysis, decision to publish, or preparation of the manuscript.

==============================
Objective

A putative relationship between markers for the serotonin system and the personality scale self-transcendence (ST) and its subscale spiritual acceptance (SA) has been demonstrated in a previous PET study of 5-HT1A receptor binding in healthy control subjects. The results could however not be replicated in a subsequent PET study at an independent centre. In this study, we performed a replication of our original study in a larger sample using Bayesian hypothesis testing to evaluate relative evidence both for and against this hypothesis.

Methods

Regional 5-HT1A receptor binding potential (BPND) was examined in 50 healthy male subjects using PET with the radioligand [11C]WAY100635. 5-HT1Aavailability was calculated using the simplified reference tissue model (SRTM) yielding regional BPND. ST and SA were measured using the Temperament and Character Inventory (TCI) questionnaire. Correlations between ST/SA scores and 5-HT1ABPND in frontal cortex, hippocampus and raphe nuclei were examined by calculation of default correlation Bayes factors (BFs) and replication BFs.

Results

There were no significant correlations between 5-HT1A receptor binding and ST/SA scores. Rather, five of six replication BFs provided moderate to strong evidence for no association between 5-HT1A availability and ST/SA, while the remaining BF provided only weak evidence.

Conclusion

We could not replicate our previous findings of an association between 5-HT1A availability and the personality trait ST/SA. Rather, the Bayesian analysis provided evidence for a lack of correlation. Further research should focus on whether other components of the serotonin system may be related to ST or SA. This study also illustrates how Bayesian hypothesis testing allows for greater flexibility and more informative conclusions than traditional p-values, suggesting that this approach may be advantageous for analysis of molecular imaging data.

Introduction

The serotonin system is involved in a wide range of fundamental physiological functions like regulation of mood, sleep and appetite (Filip & Bader, 2009). Furthermore, serotonergic neurotransmission is implicated in higher brain functions such as cognitive performance (Jenkins et al., 2016) and in several psychiatric disorders, including depression, autism, anxiety disorders and schizophrenia (Filip & Bader, 2009; Fidalgo, Ivanov & Wood, 2013).

With regard to personality, the serotonin system has been linked to the trait self-transcendence (ST) in both Positron Emission Tomography (PET) and genetic studies (Borg et al., 2003; Ham et al., 2004; Lorenzi et al., 2005; Nilsson et al., 2007; Aoki et al., 2010; Saiz et al., 2010; Kim et al., 2015). ST refers to the degree to which an individual feels part of nature and the universe at large, and to extraordinary experiences such as extra sensory perception and sense of a transcendent being or presence (Gillespie et al., 2003). The association has been interpreted as evidence for a role for the serotonin system in spiritual experiences, as well as providing a putative mechanism for the involvement of serotonin in psychosis, since high scores in ST has been linked to the schizophrenia spectrum disorders (Nitzburg, Malhotra & DeRosse, 2014).

Our group previously reported a negative correlation between 5-HT1A receptor binding potential (BPND), as measured with PET and the radioligand [11C]WAY-100635, and ST as measured using Temperament and Character Inventory (TCI). The association was strongest for the subscale spiritual acceptance (SA) (Borg et al., 2003). However, the results could not be replicated in a subsequent PET study at an independent centre (Karlsson et al., 2011). These studies contained 15 and 20 healthy participants, respectively, and therefore, a replication study in a larger sample is required.

Aims of the study

The aim of the present study was to perform a replication of our original finding of a negative correlation between 5-HT1A receptor BPND and ST/SA in a larger sample. In addition to traditional frequentist statistics, we made use of Bayesian hypothesis testing, which allows us not only to test a hypothesis, but also to quantify the relative probability of the observed data under competing hypotheses. Recently replication Bayes factors (BF) have been introduced (Verhagen & Wagenmakers, 2014; Wagenmakers, Verhagen & Ly, 2016), allowing researchers to evaluate replication success by taking the outcome of the previous study fully into account. In this way, we aimed to evaluate the relationship between 5-HT1A receptor binding and ST/SA from the perspective both of hypothesis testing without consideration of the magnitude of previous results, and of replication success.

Material and Methods

Subjects

The sample consisted of 50 healthy men: 12 were enrolled as control subjects in a series of different pharmacological studies (for details see Matheson et al. (2015); 38 in a twin study (Borg et al., 2016). Age ranged from 21 to 55 (Mean = 30, SD = 5 years). The studies were approved by the Regional Ethics Committee in Stockholm and the Radiation Safety Committee of the Karolinska Hospital, and all subjects provided written informed consent prior to their participation in the studies (IRB 2008/60-31/3; for serotonin markers 2013/136-32).

MR and PET data acquisition (5-HT1A binding potential)

Magnetic Resonance Imaging (MRI) images were acquired using a 1.5TGE Signa system (Milwaukee, WI, USA). T1- and T2-weighted MRI images were acquired for all subjects. The PET system used was Siemens ECAT Exact HR 47 (CTI/Siemens, Knoxville, TN, USA). All subjects were examined using [11C]WAY-100635; The injected radioactivity was 276 ± 35 MBq (mean; SD). BPND values were calculated for the same regions as examined in the original study (Borg et al., 2003): frontal cortex, hippocampus (using the simplified reference tissue model - SRTM) and dorsal raphe nucleus (using a wavelet-based method using the non-invasive Logan plot in order to reduce the noise in this small region). For detailed description see Matheson et al. (2015). Other regions were not included in the analysis as they were not part of the original study. However, since [11C]WAY100635 BPND is highly correlated between regions, the inclusion of more regions would therefore be unlikely to provide unique information from the three included regions (Bose et al., 2011).

Personality assessment

The Swedish translation of the TCI self-report questionnaire was used (Brändström et al., 1998). It consists of 238 true/false items covering four temperament dimensions (novelty seeking, harm avoidance, reward dependence, and persistence) and three character dimensions (self-directedness, cooperativeness, and self-transcendence). Individual scores were calculated for ST and its subscale SA.

Statistical analysis

Pearson’s correlation coefficients and their corresponding p-values were calculated for the correlation between ST/SA and 5-HT1A BPND in the frontal cortex, hippocampus and dorsal raphe nucleus. Two BF tests were performed for each comparison. Firstly, we calculated a default correlation BF for the association between BPND and the ST/SA scores in frontal cortex, hippocampus and dorsal raphe nucleus respectively. Since we specifically wanted to test a negative correlation, we choose a one-sided default Bayes factor test, with a negative Beta prior of width 1 (i.e., flat between −1 and 0) using JASP (JASP Team, 2017). This test compares the predictive adequacy of the null hypothesis H0 (i.e., no correlation) with an alternative hypothesis H- (i.e., a negative correlation) (for more details on Bayes factors, see (Ly, Verhagen & Wagenmakers, 2016; Wagenmakers, Morey & Lee, 2016). Second, we calculated a replication BF for the correlations for each region as a measure of replication success. This test compares the predictive adequacy of the null hypothesis H0 (i.e., no correlation) with an alternative hypothesis Hr. The alternative hypothesis is defined as the posterior distribution of the correlation coefficient derived from the original study, assuming a uniform prior before seeing the data of the original study (Wagenmakers, Verhagen & Ly, 2016). We slightly modified of the following source code http://www.josineverhagen.com/wp-content/uploads/2013/07/RepfunctionscorrelationFINAL1.txt (for plotting purposes) to the code which can be found online at the following address: https://osf.io/x9gjj/. This code was executed using RStudio (RStudio Team, 2017) with R 3.3.2 (R Core Team, 2015). We also reanalysed the results of Karlsson et al. (2011) with these methods. Bayes factors assess the relative likelihood of the observed data under competing hypotheses, yielding a ratio of the relative evidence for one hypothesis over the other. For instance, a BF01 below 3 indicates weak or anecdotal evidence, a BF01>3 moderate and a BF01 >10 strong evidence in favour of the null against the alternative (Jeffreys, 1961). In this paper, all BFs are presented as the likelihood of the null hypothesis relative to the alternative hypothesis (i.e., BF0− specifying a negative correlation as alternative; BF0r specifying the posterior probability distribution of the original correlation as alternative). The differences between the default and the replication BF tests can be expressed as follows: the default test addresses the question of whether an effect was present or absent given relatively little prior knowledge of the effect size, while the replication test asks whether the effect was similar to what was found before, or absent (Wagenmakers, Verhagen & Ly, 2016).

Two potential sources of bias for this analysis were the inclusion of twin pairs, and the use of cerebellar grey matter as the reference region (Hirvonen et al., 2007). We therefore performed two additional analyses by (1) randomly excluding one twin from each twin pair (using http://www.random.org), resulting in a sample size of 31, and (2) using the white matter as a reference region for hippocampus and frontal cortex.

Results

In the present sample of 50 subjects, the BPND of [11C]WAY100635 varied about 4-fold between individuals (Table 1). ST scores ranged from 2 to 24 (mean 9.7, SD 5.8); the SA scores ranged from 0 to 12 (mean 3.9, SD 3.1) (Table 1). There were no significant correlations between regional 5-HT1A receptor binding and scores on ST or SA (Fig. 1, Table 2).

Table 1 TCI scores and BPND in the original study (Borg et al., 2003) and the present replication study.

	Original study	Replication	
	Mean (SD)	Range	Mean (SD)	Range	
TCI scores					
ST	9.4 (3.8)	3–15	9.7 (5.8)	2–24	
SA	4.7 (3.0)	0–9	3.9 (3.1)	0–12	
BPND values					
Dorsal raphe nuclei	2.2 (0.87)	0.81–4.11	1.7 (0.48)	0.64–2.88	
Hippocampus	4.7 (1.49)	1.91–7.15	5.1 (1.41)	2.27–8.14	
Frontal cortex	3.2 (0.90)	1.60–4.55	3.3 (0.73)	1.21–4.61	
Notes.

Abbreviations TCI Temperament and Character Inventory

ST self-transcendence

SA spiritual acceptance

BPND binding potential

Figure 1 Correlation between self-transcendence (ST) and spiritual acceptance (SA) scales on TCI and 5-HT1A receptor binding potential (BPND) in frontal cortex, dorsal raphe nuclei and hippocampus in 50 healthy men

(A) Self-transcendence in frontal cortex. (B) Self-transcendence in hippocampus. (C) Self-transcendence in dorsal raphe nuclei. (D) Spiritual acceptance in frontal cortex. (E) Spiritual acceptance in hippocampus. (F) Spiritual acceptance in dorsal raphe nuclei. Abbreviations: TCI, Temperament and Character Inventory.

Table 2 Pearson’s r, default BF and replication BF for 5-HT1A BNND and self- transcendence/spiritual acceptance for frontal cortex, hippocampus and dorsal raphe nuclei for present & Karlsson’s replication.

	Pearson’s r	P-value	Present Replication	Karlsson Replication	
			BF0−	BF0r	BF0−	BF0r	
Self-transcendence (ST)	
- frontal cortex	−0.06	0.70	5.3	8.1	4.8	7.6	
- hippocampus	−0.19	0.20	2.5	2.3	5.0	9.0	
- dorsal raphe nuclei	−0.11	0.46	4.3	6.4	1.8	2.4	
Spiritual acceptance vs material rationalism (SA)	
- frontal cortex	−0.03	0.86	5.6	12.8	6.0	12.3	
- hippocampus	−0.12	0.41	4.1	31.5	4.0	33.8	
- dorsal raphe nuclei	−0.16	0.27	3.1	21.2	2.6	17.8	
Notes.

Abbreviations TCI Temperament and Character Inventory

r Pearson’s correlation efficient

BF0− the default BF representing the relative likelihood of the null hypothesis (H0: no correlation) compared to the alternative hypothesis (H − : negative correlation), given the data

BF0r replication BF representing the relative likelihood of the null hypothesis (H0 : no correlation) compared to the alternative hypothesis Hr obtained from the original study (Hr : posterior of ρ given the original study), given the data

All BF favoured the null over the alternative hypotheses. Default correlation BFs ranged from 2.5 to 5.6 in favour of the null (Table 2), meaning that the null hypothesis of no correlation is 2.5 to 5.6 times more likely than the alternative hypothesis for a negative correlation, given the data. For the results of Karlsson et al. (2011), default correlation BFs ranged from 1.8 to 6.0. Nine out of 12 default BFs provided moderate evidence in favour of the null hypothesis; the remaining three provided only weak evidence (Table 2).

The replication BFs ranged from 2.3 to 31.5 in favour of the null hypothesis (Table 2); replication BFs for Karlsson et al. (2011) ranged from 2.4 to 33.8. Ten out of 12 replication BFs provided moderate to strong evidence in favour of the null hypothesis. The remaining two replication BFs provided only weak evidence (Table 2).

Figure 2 illustrates the replication BF, showing how the data from the replication study shifts the distribution from the original study towards a correlation coefficient close to zero.

Figure 2 Prior and posterior probability distributions for the correlation coefficient for the Bayesian test for replication of the correlation between self-transcendence/spiritual acceptance and 5-HT1A BPND.

(A) Self-transcendence in frontal cortex. (B) Self-transcendence in hippocampus. (C) Self-transcendence in dorsal raphe nuclei. (D) Spiritual acceptance in frontal cortex. (E) Spiritual acceptance in hippocampus. (F) Spiritual acceptance in dorsal raphe nuclei. The curves represent conditional probability distributions, π, of the correlation coefficient (ρ), given the data (d) for the original (ori), replication (rep) and both studies together (all). π(ρ) represents the uniform prior distribution assumed for the original study (Borg et al., 2003). π(ρ|dori) represents the posterior distribution of the original study. π(ρ|drep) represents the posterior distribution of the replication study, assuming a uniform prior (i.e., without taking the results of the original study into consideration—the posterior based on a uniform prior on ρ). π(ρ|dall) represents the posterior distribution of the replication study using the posterior distribution of the original study as prior (i.e., the posterior distribution taking the results of both studies into consideration). The grey points indicate the height of the prior and posterior distributions at the sceptic’s null hypothesis that the effect size is zero. The ratio of these two points is the replication BF. Abbreviations: BF0r, replication BF representing the relative likelihood of the null hypothesis (H0 no correlation) compared to the alternative hypothesis Hr obtained from the original study (Hr: posterior of ρ given the original study), given the data.

The results did not greatly differ after repeating the analysis to account for biases, either by randomly excluding one twin from each twin pair, or by using white matter as reference region (see Supplemental Information 1).

Discussion

The aim of the present study was to perform a replication of our previous study (Borg et al., 2003) in a larger sample. We were not able to find any significant relationships between 5-HT1A receptor availability and ST/SA for any of the three regions. This is in line with the results of Karlsson and co-authors in an earlier replication study (Karlsson et al., 2011). Instead, in both this study and in our reanalysis of the results of Karlsson et al. (2011), Bayesian analysis provided more support for the null-hypothesis i.e., that 5-HT1A receptor is not related to the propensity for extraordinary or transcendental experiences

Despite the present results, the serotonin system remains of interest in research on the biological underpinning of personality traits associated with extraordinary experiences. 5-HTT (serotonin transporter) has been linked to ST in both a PET study (Kim et al., 2015), and in genetic studies—though results are conflicting (Nilsson et al., 2007; Aoki et al., 2010; Saiz et al., 2010). Furthermore, 5-HT1A,5-HT2A and 5-HT6 receptor gene polymorphisms have been shown to be correlated to ST (Ham et al., 2004; Lorenzi et al., 2005).

Pharmacological research shows that the serotonin system plays a key role in the effects of hallucinogens, which produce psychosis-like symptoms (comparable to some of the items in the SA scale) (Vollenweider et al., 1999; Geyer & Vollenweider, 2008). Moreover, treatment with SSRI in depressed patients lowered ST scores (Hruby et al., 2009).

Hence, although we failed to replicate the association between 5-HT1A and ST/SA, these lines of evidence motivate further research to clarify the role of serotonin neurotransmission and ST/SA in the healthy population as well as in patients.

The present study was performed on an independent sample of healthy male individuals. Compared to our original study, the sample exhibited less variance in age, and 38 of the 50 subjects were twin pairs. TCI scores and BPND values were however similar to the original study, therefore the more homogenous age range and genetic background of the present sample are unlikely to fully explain the difference in results. Furthermore, we used more advanced image processing methods than in our original study (although many of these, such as automated region of interest (ROI) definition and frame-by-frame realignment of the PET images, were also used in the study by Karlsson and colleagues (Hirvonen et al., 2008; Karlsson et al., 2011). We were not able to reanalyse the data of the original study using these methods, since T1 weighted MR images were not collected in this sample. However, automated ROIs have been shown to exhibit similar reliability compared to manual (Johansson et al., 2016), suggesting that methodological factors are unlikely to explain the discrepancies.

Replication failure is a common problem in science: in clinical trials and psychology studies replication rates range from 11 to 39%, respectively (Begley & Ellis, 2012; Open Science Collaboration, 2015). Both previous studies on 5-HT1A and ST/SA had low power due to small sample sizes and multiple comparisons without correction, possibly leading to incorrect inferences. According to our calculations using PPV (positive predictive value; the probability that a ‘positive’ research finding reflects a true effect) (Button et al., 2013) the probability that our original finding was true was only around 9%, even before consideration of the two replication studies (see Supplemental Information 1 for the assumptions and the calculation).

Limitations

Our data consisted of males only. We excluded women from the analysis since the literature is conflicting about the effect of gender and menstrual cycles on 5-HT1A receptor binding (Palego et al., 1997; Tauscher et al., 2001; Cidis Meltzer et al., 2001; Parsey et al., 2002; Costes et al., 2005; Jovanovic et al., 2008; Stein et al., 2008; Moses-kolko et al., 2011) and gender influences ST scores on TCI (Brändström, Richter & Przybeck, 2001; Garcia-Romeu, 2010). Additionally, we wanted to replicate our original study, which contained only males, as closely as possible. Therefore, caution must be exercised when generalizing the present finding in male subjects to the female population. Karlsson and co-authors studied a gender mixed sample (11 males/nine females) in their previous negative study (Karlsson et al., 2011), and in genetic studies the association between serotonin genes and ST/SA has in some studies been reported to differ between gender (Nilsson et al., 2007; Aoki et al., 2010) whereas others found no difference (Lorenzi et al., 2005; Saiz et al., 2010).

The same is true for age: we had a similar sample to the original study, with limited range, and age might influence both ST scores and 5-HT1A binding (Kirk, Eaves & Martin, 1999; Brändström, Richter & Przybeck, 2001; Moses-kolko et al., 2011).

As in the original study, we used the cerebellar grey matter as a reference region, which is not considered the gold standard due to small levels of specific binding in this region (Shrestha et al., 2012). However, using arterial plasma to calculate BPP and BPND using cerebellar white matter as reference, Karlsson and co-authors could not replicate the original findings either (Karlsson et al., 2011). In addition, our analysis using cerebellar white matter showed similar results (see Supplemental Information 1).

Strengths

Where Karlsson and co-authors could only conclude that they did not find a significant correlation between ST/SA and 5-HT1A receptor binding (Karlsson et al., 2011), using Bayesian hypothesis testing, we were able to conclude that the data supplied more evidence in favour of the null hypothesis (i.e., no correlation) for both our data and for that of Karlsson et al. (2011). Furthermore, the replication BF allowed us to take the magnitude of our previous results and its uncertainty fully into account. In this way, using the current data, the replication BF results suggest that the effect reported by the original study was likely either to be overestimated or a false positive. As such, these results support the conclusion that there is little to no association between ST/SA and 5-HT1A receptor binding.

Of wider interest in the field of molecular imaging is that Bayesian hypothesis testing provides more informative conclusions than traditional p-values, thus offering pragmatic advantages for analysis of expensive neuroimaging studies, where limited sample sizes are common. For instance, Bayesian hypothesis testing allows for collecting data until the evidence is sufficiently strong to make a conclusion for one or the other hypothesis without requiring correction for multiple comparisons with sequential analyses. In this way, both costs and radiation exposure can be decreased.

Conclusions

In conclusion, we failed to replicate our previous finding of a negative association between ST/SA and 5-HT1A receptor binding. Rather, our Bayesian analysis found more evidence for a lack of correlation. Further research should focus on whether other components of the serotonin system may be related to ST/SA.

Supplemental Information

Supplemental Information 1 Supplementary information

Click here for additional data file.

We gratefully thank the members of the PET group at the Karolinska Institutet for assistance over the course of the investigation.

Additional Information and Declarations

Competing Interests

Author Contributions

Human Ethics

Data Availability

Lars Farde is employed by AstraZeneca Pharmaceuticals. Simon Cervenka has received grant support from AstraZeneca as co-investigator, and has served as a one off speaker for Roche and Otsuka Pharmaceuticals.

Gina Griffioen analyzed the data, contributed reagents/materials/analysis tools, prepared figures and/or tables, authored or reviewed drafts of the paper, approved the final draft.

Granville J. Matheson performed the experiments, contributed reagents/materials/analysis tools, authored or reviewed drafts of the paper, approved the final draft.

Simon Cervenka authored or reviewed drafts of the paper, approved the final draft.

Lars Farde conceived and designed the experiments, contributed reagents/materials/analysis tools, authored or reviewed drafts of the paper, approved the final draft.

Jacqueline Borg conceived and designed the experiments, performed the experiments, authored or reviewed drafts of the paper, approved the final draft.

The following information was supplied relating to ethical approvals (i.e., approving body and any reference numbers):

The studies were approved by the Regional Ethics Committee in Stockholm and the Radiation Safety Committee of the Karolinska Hospital, and all subjects provided written informed consent prior to their participation in the studies (IRB 2008/60-31/3; for serotonin markers 2013/136-32).

The following information was supplied regarding data availability:

Unfortunately, due to institutional refusal to share data openly based on Swedish national law, we can only publish our metadata openly: Griffioen G. 2018. “Serotonin 5-HT1A Receptor Binding and Self-Transcendence in Healthy Control Subjects - a Replication Study Using Bayesian Hypothesis Testing.” OSF. April 4. http://osf.io/x9gjj.

The underlying data are pseudonymised according to national (Swedish) and EU legislation, and can’t be anonymised and published in an open repository. The data can instead be made available upon request on a case by case basis as allowed by the legislation and ethical permits. Requests for access can be made to the Karolinska Institutet’s Research Data Office at http://rdo@ki.se.

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
