# Peer review of "Serotonin 5-HT1A receptor binding and self-transcendence in healthy control subjects—a replication study using Bayesian hypothesis testing"

_PeerJ, doi:10.7717/peerj.5790_

## Round 0.1 · original submission · Major Revisions

I am the Academic Editor who has been asked to handle your Appeal. I have reviewed the prior materials and reviews and believe there is a case to be made for a new round of review/decision. I am thus rendering this "revision" decision to allow you to upload the new version of the submission that you have prepared. I will then re-review that version once it is submitted.

· Appeal

Appeal


· · Academic Editor

Reject

Dear Authors, Unfortunately Due to the comments received the manuscript is not ready to be published in PeerJ.We hope that you and your team will continue to submit new manuscripts to Peer J in the coming future.

# Staff Note: This decision was reviewed, and approved by a PeerJ Section Editor, covering this section in the journal

Reviewer 1 ·

Basic reporting

Identical work is put online https://www.biorxiv.org/content/early/2017/12/22/226092. Can authors explain?


No caption in all the tables and figure.

Experimental design

This work is a replication of the previous work from the same group that already published in 2003. The number of subjects has increased from 15 to 50. The quantity is sufficient to make the statistical sound, but the selection of subject is still considered too narrow with only single gender in certain age group. Bayesian approach is used here to proof the correlation between the serotonin binding and self-transcendence. It was concluded with no correlation was found. In general, the study is too vague to jump into such conclusion. Bayesian approach in biological or medical used is quite a common approach in modelling. But thorough experimental design and details questionnaire will be needed in order to to test the hypothesis and make a reliable conclusion. Self-transcendence is very subjective. Can authors defined self-transcendence since it is a very subjective behavior ? How authors define the scoring for self transcendence?

Validity of the findings

No comment .

Comments for the author

The current form of the work is not ready for publication.

·

Basic reporting

The writing is good. Unfortunately, it was not clear to me what an inverse correlation is. Do the authors mean a negative correlation? If so, please just write negative correlation, otherwise please clarify.

Experimental design

No idea about the Aims and Scope of the journal, but it was clear that the authors tried to replicate their previous result as closely as possible, which is admirable.

Validity of the findings

As is written on line 128, an unaware reader might misinterpret the replication hypothesis, where it is stated that H_{r} is the original correlation. It’s true that Josine’s R-code uses the original observed correlation r_{ori} and n_{ori}, but it does not mean that the alternative hypothesis is specified as the point r_{ori}. Instead, a whole range of population correlation rho’s is specified as the alternative hypothesis, see Figure 2 (in fact, the code provided in this review). This aforementioned whole range is the posterior on rho informed by the data of the original study. It is correct that this posterior is peaked at r_{ori} and its concentration depends on n_{ori}; the larger n_{ori} the more peaked it is. As n_{ori} is not extremely large it is definitely not a point hypothesis. In other words, the replication Bayes factor uses the posterior based on data of the original study as a prior for the replication attempt. Moreover, in Ly, Etz, Marsman, & Wagenmakers (2017), see

https://psyarxiv.com/u8m2s/

it is clarified that the replication Bayes factor quantifies the evidence within the data of the replication attempt given the data of the original study. Perhaps the following references might help the reader with the interpretation of the Bayes factor. Though that’s fully up to the authors.

Jeffreys, H. (1961). Theory of Probability

Ly, A., Verhagen, A. J., & Wagenmakers, E.-J. (2016). Harold Jeffreys’s default Bayes factor hypothesis tests: Explanation, extension, and application in psychology.

Ly, A., Verhagen, A. J., & Wagenmakers, E.-J. (2016). An evaluation of alternative methods for testing hypotheses, from the perspective of Harold Jeffreys.

Comments for the author

Unfortunately, I was unable to access the data, as the file

~/Dropbox/PET studie/NenR orig&repl.xlsx

isn’t uploaded onto the OSF. It would be nice to have this available as I would then be able correct Figure 2 of the manuscript. The following script requires the author to download (today 01-03-2018)

https://github.com/AlexanderLyNL/jasp-desktop/blob/development/JASP-Engine/JASP/R/common.R
https://github.com/AlexanderLyNL/jasp-desktop/blob/development/JASP-Engine/JASP/R/correlationbayesian.R

and source them. I highly recommend putting “common.R” and “correlationbayesian.R” on the OSF repository, as the code will be modified in the near future. I’m not fond of the name “default posterior”, but it’s up to the authors. Lastly, as the authors have the original and the replication data at hand, they can much easier calculate a replication Bayes factor, perhaps after a location shift. I’m quite sure that the authors of Ly, Etz, Marsman, & Wagenmakers (2017) are willing to share this code.

myWd <- "~/Dropbox/PET studie/"
setwd(myWd)

source("common.R")
source("correlationbayesian.R")

# Fill in the original and replication result
nOri <- 12
rOri <- -0.06

nRep <- 50
rRep <- -0.19

bfObject <- .bfCorrieRepJosine(nOri=nOri, rOri=rOri, nRep=nRep, rRep=rRep)

rComb <- bfObject$combined$r[2]
nComb <- bfObject$combined$n[2]


myFirstPriorLine <- .priorRhoMin(rho=rhoDomain)
myPosteriorGivenDOriLine <- .posteriorRhoMin(n=nOri, r=rOri, rho=rhoDomain)
mySecondPriorLine <- myPosteriorGivenDOriLine
myPosteriorGivenAll <- .posteriorRhoMin(n=nComb, r=rComb, rho=rhoDomain)

maxY <- 1.05*max(myFirstPriorLine, myPosteriorGivenDOriLine, myPosteriorGivenAll)


par(cex.main=3, mgp=c(3.5, 1, 0), cex.lab=2.5, font.lab=2, cex.axis=1.3,
mar=c(5, 6, 4, 7)+0.1, bty="n", las=1)
plot(0, 0, xlab=expression(rho), axes=FALSE, ylab="", ylim=c(0, maxY), col="white")
axis(1)
axis(2)
lines(rhoDomain, myFirstPriorLine, lwd=3, lty=3, col="grey")
lines(rhoDomain, mySecondPriorLine, lwd=3, lty=2, col="dimgrey")
lines(rhoDomain, myPosteriorGivenAll, lwd=3)

points(0, mySecondPriorLine[zeroIndex], pch=21, cex=2.5, bg="grey")
points(0, mySecondPriorLine[zeroIndex], cex=2.5, lwd=3)

points(0, myPosteriorGivenAll[zeroIndex], pch=21, cex=2.5, bg="grey")
points(0, myPosteriorGivenAll[zeroIndex], cex=2.5, lwd=3)

legend("topright", c(expression(paste(pi["-"], "(",rho, ")")),
expression(paste(pi["-"], "(",rho, "|d"['ori'], ")")),
expression(paste(pi["-"], "(",rho, "|d"['comb'], ")"))),
lwd=c(2, 2, 2), lty=c(3, 2, 1), col=c("grey", "dimgrey", "black"),bty='n', cex=1.3)

---

## Round 0.2 · Minor Revisions

I agree that the approach you have taken here to revisit your previous results, and to quantify the evidence pro and con your original hypothesis, is an important example for the field. Your response to the reviews was thorough (and exceptionally patient). **However,** please do edit the manuscript to respond to the minor changes suggested by reviewer 2. A detailed response letter is not needed.

In addition, the Editorial Office has emailed you separately about your GDPR concern. Once that issue is resolved I will be happy to Accept your manuscript and in the meantime I am issuing this Minor Revision decision while you discuss at your end.

·

Basic reporting

The conclusion in the previous round to reject the paper was and is, in my opinion, unfair and preposterous. As a result of this, I decided not to review for this journal again. Fortunately, the authors appealed and I’m glad that they got the opportunity to resubmit the manuscript, which made me decide to review again. The current manuscript addresses all the concerned raised previously and I believe that it should be published.

Experimental design

Good

Validity of the findings

Good

Comments for the author

Very minor remarks:
- Line 76: Please remove "a" and make it plural. "Recently replication Bayes factors"
- Line 123: Please write Bayes factors instead of Bayes Factors.
- Line 128: It would be more clear if the authors would write "... original study, assuming a uniform prior before seeing the data of the original study"
- Line 136: Please add the subscripts to the Bayes factor and elaborate that the Bayes factor is a relative measure of evidence. For instance "A BF01 below 3 indicates weak or anecdotal evidence...... strong evidence in favour of the null against the alternative."
- Line 137: Etz & Vandekerckhove, 2016 are great, but Harold Jeffreys (1961) should be credited for this.
- Line 158: Before this paragraph it might be good to give the overall conclusion that all Bayes factor favour the null over the alternatives.
- Line 252: Is a bit unclear to me
- Table 2 description:
+ H_{r}: Posterior of \rho given the original study
+ In their response the authors noted that they did not do a directional test, but the notation BF01 remained. In addition, my personal preference is denote a replication Bayes factor as BF01(d_{\rep} | d_{\orig}) to make explicit that one conditions on the previous data. The code I provided gives Bayes factors and replication Bayes factors for both directed as well the undirected tests. Thanks for spotting the error in my code.
- Figure 2 description:
+Please change "The posterior of the default BF test" to "The posterior based on a uniform prior on \rho".
+H_{r}: Posterior of \rho given the original study

---

## Round 0.3 · accepted · Accept

Please provide a statement in the paper about the institutional refusal to share data openly and the conditions for data availability (e.g., must sign an agreement, or whatever).

# #